# Prevalence and Risk Factors of Depression in Patients with Rheumatic Disease in South Korea during the COVID-19 Pandemic

**DOI:** 10.3390/healthcare10091758

**Published:** 2022-09-13

**Authors:** Sanghyun Bae, Ok-Hee Cho

**Affiliations:** 1Department of Nursing, Shinsung University, Dangjin-si 31801, Korea; 2Department of Nursing, College of Nursing and Health, Kongju National University, Gongju-si 32588, Korea

**Keywords:** COVID-19, depression, insomnia, post-traumatic stress disorder, rheumatic diseases

## Abstract

This study aimed to examine the prevalence and risk factors of depression among patients with rheumatic diseases (RDs) during the coronavirus disease 2019 (COVID-19) pandemic. This study adopted a cross-sectional design, and 160 outpatients with RDs in one university hospital in South Korea were sampled using the convenience sampling method. Data were collected from May to July 2021 using a structured questionnaire. The risk factors of depression were analyzed using descriptive statistics and multiple logistic regression analyses. The prevalence rates of post-traumatic stress disorder (PTSD), insomnia, and depression were 37.5%, 20.0%, and 24.4%, respectively. Multiple logistic regression analyses confirmed that employment status, monthly income, perceived health, PTSD, and insomnia were significant risk factors of depression. The findings highlight the urgent need to assist patients with RDs who are at risk of depression during the COVID-19 pandemic, especially individuals who are unemployed or have low incomes and poor perceived health, individuals with high PTSD, and individuals with severe insomnia. There is a need to provide disease-specific interventions to effectively alleviate depression among these individuals during the COVID-19 pandemic.

## 1. Introduction

The coronavirus disease 2019 (COVID-19) remains active and continues to spread worldwide. Despite aggressive countermeasures such as reinforced anti-infection systems, vaccination campaigns, and development of therapeutics, uncertainties engendered by the prolonged pandemic pose grave threats to societies and economies [1]. As of May 2022, more than 531.19 million patients and 6.314 million deaths have been recorded worldwide, and approximately 18.104 million patients and 25,000 deaths have been recorded in South Korea [2]. Previous studies have identified old age, obesity, hypertension [3], diabetes mellitus, cardiovascular disease, chronic obstructive pulmonary disease, end-stage renal disease, cancer, and autoimmune diseases as the risk factors for severe complications and death owing to COVID-19 [4]. Of them, rheumatic diseases (RDs; e.g., rheumatoid arthritis (RA), ankylosing spondylitis, systemic lupus erythematosus) particularly hinder the detection of a COVID-19 infection because the manifestation of RDs include systemic symptoms such as chronic inflammation and multi-organ impairments. In addition, these patients are immunocompromised owing to prolonged use of steroids and immunosuppressants. Therefore, these patients may be more vulnerable to COVID-19 infection and complications than healthy adults [5]. The Korean Ministry of Health and Welfare raised the level of emergency alert for infectious diseases due to the spread of COVID-19 to “severe” in February 2020, which was maintained until May 2022. During the data collection period of this study (May–July 2021), the spread of infection surged again due to the appearance of new strains, including the delta mutation in Korea, and it was recommended to refrain from using multi-use facilities where a large number of people gather [6].

During the current pandemic, the prevalence of depression has been higher among the RD population than in the general population [7]. As patients with RDs must adapt to their disease and comply with treatment regimens for the rest of their lives, depression is a common mental health disorder observed in this population owing to increased disease activity, suboptimal treatment adherence, reduced treatment response, and decreased quality of life [8,9]. Fear of health risk, stress, social isolation [10], anxiety [11], disease activity [12], physical symptoms [13], and insomnia [14] during the current pandemic can elevate the risk of depression or exacerbate depression symptoms among patients with RDs. Prior studies have reported that the unprecedented changes in the social environment caused by social distancing, restricted social gatherings, working from home, online-centered lifestyle, and quarantine during the pandemic posed serious mental health threats to patients with RDs. This is because they require routine medical attention; therefore, restricting their access to healthcare [15] thus induces post-traumatic stress disorder (PTSD) [16] and triggers insomnia [17]. However, research on the effects of the characteristics of patients with RDs (perceived health status, social isolation, PTSD, and insomnia) on the prevalence of clinical depression amid a prolonged pandemic is lacking. In this context, we examined the level of depression, its associated factors, and the association between these factors to identify the risk factors of depression among patients with RDs during the COVID-19 pandemic. We hope to present valuable baseline data for developing effective prevention interventions by enabling the early identification of patients at risk for depression.

## 2. Materials and Methods

### 2.1. Study Design

This study utilized a cross-sectional, correlational survey to identify the risk factors of depression among patients with RDs during the COVID-19 pandemic.

### 2.2. Participants and Data Collection

The entire study protocol was approved by the Institutional Review Board at the Daejeon Eulji Medical Center (No. EMC202011008). Data were collected from May to July 2021. Participants were convenience sampled from the outpatients of the RD department at one university hospital in South Korea. The inclusion criteria were being aged ≥ 19 years and currently receiving outpatient care owing to RA, systemic lupus erythematosus, or ankylosing spondylitis. The exclusion criteria were current acute treatment for RDs, diagnosis of a mental disorder, and use of anxiety or depression medications. Regarding sample size, as 10 participants are needed per one predictor variable (i.e., the independent variable) [18], the minimum sample size required for binomial logistic regression analysis for 13 study variables was 130. After explaining the purpose and method of the study to the eligible participants, the questionnaire was distributed to those who signed informed consent. A total of 163 participants were initially recruited, and after excluding three questionnaires with invalid responses, data from 160 participants were included in the final analysis. The participants completed the questionnaires independently, and the questionnaires took about 20 min to complete. A small gift was offered to the participants who completed the questionnaires.

### 2.3. Instruments

#### 2.3.1. General Characteristics

Altogether, 10 items were used to survey general characteristics: age, sex, marital status, education level, employment status (yes/no), monthly income, diagnosis, time since diagnosis, perceived health (poor/moderate/good), and perceived social isolation. Perceived social isolation was assessed using the following question: “In the past two weeks, how much have you felt socially isolated or lonely owing to COVID-19?” Participants were asked to choose a number on a visual analog scale from 0 (“none”) to 10 (“extreme”). A higher score indicated greater perceived social isolation owing to COVID-19.

#### 2.3.2. PTSD

PTSD was assessed using the Korean version [19] of the Revised Impact of Event Scale (IES-R), developed by Horowitz et al. [20] and modified by Weiss et al. [21]. This 22-item scale is rated on a five-point Likert scale ranging from 0 (“not at all”) to 4 (“very much”). The total score ranges from 0 to 88, and a higher score indicates more severe PTSD symptoms. We used a cutoff score of 22 with reference to a previous study that validated the instrument on the South Korean population [19]. Cronbach’s α was 0.94 in this study.

#### 2.3.3. Insomnia

Insomnia severity was assessed using the Korean version [22] of the Insomnia Severity Index (ISI) developed by Morin et al. [23]. This 22-item scale is rated on a five-point Likert scale ranging from 0 (“not at all”) to 4 (“very”). The total score ranges from 0 to 28, and a higher score indicates more severe insomnia. We used a cutoff score of 15 for clinically significant insomnia with reference to a previous study that validated the instrument on the South Korean population [22]. Cronbach’s α was 0.94 in this study.

#### 2.3.4. Depression

Depression was assessed using the Korean version [24] of the Patient Health Questionnaire-9 (PHQ-9) developed by Spitzer et al. [25]. This nine-item scale is rated on a four-point Likert scale ranging from 0 (“not at all”) to 4 (“nearly every day”). The total score ranges from 0 to 27, and a higher score indicates more severe depression symptoms. We used a cutoff score of 10 (<10 for no depression, ≥10 for depression) with reference to a previous study that validated the instrument on the South Korean population [26]. Cronbach’s α was 0.90 in this study.

### 2.4. Data Analysis

The data were analyzed using the SAS 9.4 (SAS Institute Inc. Cary, NC, USA) software. The general characteristics and level of study parameters were analyzed with descriptive statistics. The differences in depression according to general characteristics were analyzed using a *t*-test and *x*^2^ tests; the effect sizes (Cohen’s d and Cramer’s V) were also evaluated. The risk factors of depression were identified using multiple logistic regression analyses.

## 3. Results

### 3.1. General Characteristics and Study Parameters

Table 1 shows the participants’ general characteristics. The mean perceived social isolation score was 4.43 ± 3.36. The mean PTSD score was 18.38 ± 14.51, and 60 had PTSD (IES-R ≥ 22). The mean insomnia score was 9.83 ± 6.42, and 32 (20.0%) had insomnia (ISI ≥ 15). The mean depression score was 6.56 ± 5.81, and 39 (24.4%) had depression (PHQ-9 ≥ 10) (Table 1).

### 3.2. Prevalence of Depression According to General Characteristics

The depression group was older than the non-depression group (*p* < 0.001). The prevalence of depression was higher among those with a high school diploma than among those with a college degree or higher (*p* = 0.034); among individuals who were unemployed than among those who were employed (*p* = 0.003); among those with a monthly income of <2 million Korean won (KRW) than among those with an income of ≥2 million KRW (*p* < 0.001); and among those with poorer (vs. better) perceived health (*p* = 0.018). The level of perceived social isolation was higher among those with depression than those without depression (*p* < 0.001). In terms of effect size, the difference was trivial for age, education level, occupation, and perceived health status and large for perceived social isolation (Table 2).

### 3.3. Risk Factors of Depression

To identify the risk factors of depression, multiple logistic regression analyses were performed with PTSD, insomnia, and the general characteristics that significantly differed in relation to depression in the univariate analysis—age, education level, employment status, monthly income, perceived health status, and perceived social isolation—as the independent variables. The results showed that employment status (OR = 3.61, 95% CI 1.02 to 12.79; *p* = 0.047), monthly income (OR = 3.92, 95% CI 1.04 to 14.80; *p* = 0.044), perceived health (OR = 16.93, 95% CI 1.93 to 14.81; *p* = 0.011), PTSD (OR = 1.10, 95% CI 1.05 to 1.16; *p* < 0.001), and insomnia (OR = 1.26, 95% CI 1.11 to 1.42; *p* < 0.001) were significant risk factors for depression (Table 3).

## 4. Discussion

This study investigated the prevalence and risk factors of depression among patients with RDs during the COVID-19 pandemic to support the development of efficient measures to promote effective coping and adaptation of patients with RDs during the prolonged COVID-19 pandemic. The prevalence of depression among patients with RDs was 24.4%, and the prevalence of depression among patients with RDs reported by previous studies since the declaration of the pandemic ranged from 8.6 to 60.9% [5,9,27]. The variations in the rate of depression reported across studies may be attributed to the differences in the regions and times of study, participant age [28], specific diagnosis of RDs, and study instruments. Further, even if the same instrument was used, variations may exist owing to the use of different cutoffs depending on sociocultural background.

The fact that approximately 25% of our study sample had clinically significant depression, despite the sample consisting of patients with RDs without a history of depression or current use of relevant drugs, highlights the urgency of interventions in this population. Itaya et al. [11] reported that the level of depression among patients with RA during the pandemic had not deviated much from the pre-pandemic period, and Johnstone et al. [8] reported that the pandemic had an impact on anxiety and quality of life of patients with RD but not on their depression. However, Johnstone et al. [8] suggested the possibility of increased risk for depression with the prolonging of the pandemic by showing that the number of patients suspected to be depressed rose markedly since the outbreak of the COVID-19 pandemic. Depression is chronic among patients with RDs, and it causes them to react more sensitively to physical pain and fatigue and may ultimately deteriorate their quality of life in the long term [29]. Furthermore, as depression can lead to more critical problems such as job loss and suicide [30,31] healthcare providers must be aware of and pay more attention to potential, current, and exacerbated depression and its associated factors during the pandemic.

Patients with RDs who were unemployed and had a lower monthly income were at a higher risk for depression than those who were employed and had a higher monthly income. Previous reports found that temporary unemployment, decrease in monthly income, and drug discontinuation owing to the pandemic were associated with reduced quality of life among patients with RDs [10] and that patients with RDs with higher anxiety due to reduced income since the pandemic experienced higher levels of stress [14]. This indirectly supports the argument that patients who are unemployed and have a lower income may be more vulnerable to depression.

We cannot conclude whether the cause of unemployment or low monthly income is owing to the pandemic because we collected cross-sectional data one year after the outbreak of the pandemic. However, in a longitudinal study with 133 young adults with RD (mean age: 28.9 years, 82% women), Jetha et al. [27] reported that the rate of employment dropped from 86% to 71% since the pandemic compared to the pre-pandemic period. Moreover, most patients reported that the pandemic has had an impact on their working conditions (92%) and occupational health and safety (74%). This supports the possibility of the pandemic influencing the work environment and income among patients with RDs. Therefore, employment-facilitating strategies and financial support measures should be planned considering occupational characteristics, so that a change in employment status and financial hardship caused by the pandemic do not lead to an exacerbation of mental health in patients with RDs. On the other hand, in the results of the univariate analysis of this study there was a difference in the occurrence of depression according to occupation and income categories, but the effect size was small (approximately 0.2–0.3). Effect size should be considered when interpreting the results as it provides the basis for determining the significance of the study results. In order to further clarify the relationship between these characteristics and the occurrence of depression, repeated studies with an enlarged sample are needed.

The risk of depression was higher among patients with RDs with poorer perceived health as compared to their counterparts. This is similar to the reports that self-perceived worsening of the RD was a risk factor for depression during the pandemic [7]; that patients with RDs with comorbidities showed a higher rate of adverse mental health [5]; and that exacerbation of disease activities was associated with psychological status such as suicidal ideation and the ability to cope with trauma, anxiety, and depressive symptoms in patients with ankylosing spondylitis [12]. Ziadé et al. [32] reported that people with poorer physical health have more severely restricted activities outside their homes, isolate themselves more, and display more severe symptoms of depression, suggesting that patients with poorer perceived health are at a higher risk for depression than their counterparts. Studies published since the outbreak of the pandemic showed that many patients with RDs (45.6%) believe they cannot continue exercising, had increased pain (75.6%), had poor well-being (49.0%), had worsened health during the lockdown (46.6%) [9], and had deteriorated health owing to reduced physical activities and increased sedentary lifestyles [33].

Changes in self-care practices during the pandemic may have contributed to declining perceived health and the development of depression. During the pandemic, depression has a greater impact on distress among individuals with chronic conditions than in the general population [34] and is associated with self-care noncompliance and reduced stress coping ability [35,36]. These findings demonstrate that healthcare providers should be aware that patients with poor perceived health are potentially at high risk for depression. In preparation for the post-COVID era and future potential pandemics, ways of utilizing various non-face-to-face platforms should be explored to provide the latest information about evidence-based self-care, infection management, prevention, and coping with psychological crises for patients.

Our findings show that PTSD is a risk factor for depression. Although our findings do not support a causal relationship between PTSD and depression, they can be understood in the same context as previous reports of a positive correlation between PTSD and depression [37,38]. In our study, the prevalence of PTSD among patients with RDs was 37.5%; that is, one out of three patients had clinically significant PTSD. In the early days of the pandemic, patients with RDs experienced anxiety and fear owing to unprecedented changes in their access to healthcare services and canceled or delayed physicians’ appointments owing to the explosive increase in COVID-19 cases [4], which might have contributed to the onset of PTSD. As the pandemic persists, patients with RDs are still flustered with changing anti-infection measures, lack of information about self-care, and uncertainties about the future [39,40]. In addition, being classified as a “vulnerable group” might have elevated their risk of depression because of inordinate restrictions from social activities and social stigmatization [5]. Liu et al. [28] also reported in their study on 898 young adults (18–30 years) that a high level of loneliness, COVID-19-specific worry, and low distress tolerance were associated with clinical depression.

According to our findings, the rate of insomnia among patients with RDs was 20.0%, and the risk of insomnia increases with the increasing severity of depression. Insomnia is common among patients with RDs owing to pain, mood changes, disease severity, and increased cytokine levels [41]. Kwiatkowska et al. [42] found that patients with RDs with severe insomnia also had severe depression, and Adnine et al. [7] reported that being a man, having a low socioeconomic status, and having RA were risk factors of major insomnia. Other studies [17,43] also showed that a prolonged pandemic could elevate the risk of depression by exacerbating sleep disorders. Ingegnoli et al. [14] reported in their study on 375 patients with RDs that 65.6% had trouble staying asleep, 63.5% had trouble falling asleep, and 29.9% had dreams about the pandemic, suggesting that depression interventions must address sleep for effective alleviation of depression in this patient group.

Despite the progression of the COVID-19 pandemic, the world is slowly reverting to pre-pandemic norms. Patients with RDs now have the experience of adapting to a pandemic, and they may have developed their own coping strategies [44]. Tee et al. [3] reported that providing health information was a protective factor for mental health during the pandemic, and Brady et al. [45] reported that light physical activity and walking were associated with reduced mental or physical fatigue and depressive symptoms during the lockdown in patients with RA, suggesting that physical activity may help promote mental health. Patients with RDs must remain in periodic care for the rest of their lives; consequently, they were vulnerable to depression even before the pandemic and have been more vulnerable to depression since the outbreak of the pandemic. We recommend that healthcare providers screen patients to identify those at high risk of depression based on our findings and aggressively intervene such as by referring them to psychological counseling.

This study had a few limitations. First, the sample was convenience sampled from patients with RDs admitted to one university hospital in South Korea, and sampling was performed with voluntary consent; therefore, the findings must be generalized with caution. Second, this study was a cross-sectional study, and patients’ psychological states before the pandemic were not evaluated. Hence, the causation between depression and its associated factors cannot be drawn. Third, the data are subject to a recall bias, as the patients were asked to recall their perceived depression and associated factors in the past two weeks. Fourth, the lack of a disease control group poses a limitation. Fifth, we could not control for a combination of two or more risk factors of depression (e.g., having insomnia and PTSD).

Despite these limitations, this study is meaningful because reliable data were collected through a one-on-one in-person interview during the pandemic and it presented evidence for identifying patients at high risk for depression by shedding light on the rate of clinically significant depression and their risk factors. Expanded evidence on depression and its risk factors in patients with RD may be useful in community public health policy and clinical decision-making to prevent progression to severe PTSD and suicide, and to support mental health care for patients with chronic diseases. We recommend researchers conduct randomized prospective, longitudinal studies to establish a causal relationship between depression and its associated factors. Moreover, we recommend that future studies examine the effects of changes in socioeconomic conditions, healthcare environments, and resources as a result of the pandemic, as well as personal characteristics (e.g., resilience and self-efficacy) on depression and that interventions to alleviate depression be developed and evaluated.

## 5. Conclusions

This study examined the rate of clinically significant depression, PTSD, and insomnia among patients with RDs at about two years after the outbreak of COVID-19. Further, we confirmed that the risk of depression increased among individuals who were unemployed and those with decreasing monthly income, decreasing perceived health, increasing PTSD, and increasing severity of insomnia. The findings serve as useful evidence for planning effective psychological support and interventions for patients with RDs to prevent and alleviate their depression. In preparation for current and future changes brought upon by the pandemic, healthcare providers may wish to provide reliable information and disease-specific interventions for patients with RDs to assist in their positive coping with and recovery from mental health threats.

## Figures and Tables

**Table 1 healthcare-10-01758-t001:** General characteristics and research variables of participants (*N* = 160).

Characteristics		n (%)	M ± SD
Age (year)			46.7 ± 13.7
Gender	Male	63 (39.4)	
	Female	97 (60.6)	
Marital status	Unmarried	48 (30.0)	
	Married	112 (70.0)	
Education level	High school and under	79 (49.4)	
	College and higher	81 (50.6)	
Occupation	Unemployed	48 (30.0)	
	Employed	112 (70.0)	
Monthly income (10,000 KRW)	<200	41 (25.6)	
	≥200	119 (74.4)	
Diagnosis	Rheumatoid arthritis	61 (38.1)	
	Systemic lupus erythematosus	44 (27.5)	
	Ankylosing Spondylitis	55 (34.4)	
Duration since diagnosis (year)	<10	65 (40.6)	
	≥10	95 (59.4)	
Subjective health status	Poor	13 (8.1)	
	Moderate	73 (45.6)	
	Good	74 (46.3)	
Perceived social isolation (VAS)			4.43 ± 3.36
PTSD (IES-R)	<22	100 (62.5)	18.38 ± 14.51
	≥22	60 (37.5)	
Insomnia (ISI)	<15	128 (80.0)	9.83 ± 6.42
	≥15	32 (20.0)	
Depression (PHQ)	<10	121 (75.6)	6.56 ± 5.81
	≥10	39 (24.4)	

M = mean; SD = standard deviation; KRW = Korean won; VAS = visual analogue scale; PTSD = post-traumatic stress disorder. IES-R = impact of events scale-revised; ISI = insomnia severity index; PHQ = patient health questionnaire.

**Table 2 healthcare-10-01758-t002:** Prevalence of PTSD, insomnia, and depression by general characteristics (*N* = 160).

Characteristics	Without Depression (n = 121)	With Depression (n = 39)	x^2^ or *t*	*p*	Effect Size
Age (year), Mean ± SD	45.5 ± 13.5	50.5 ± 14.1	−5.43	<.001	0.367
Gender					
Male	52 (82.5)	11 (17.5)	2.70	0.101	0.130
Female	69 (71.1)	28 (28.9)			
Marital status					
Unmarried	37 (77.1)	11 (22.9)	0.08	0.779	0.022
Married	84 (75.0)	28 (25.0)			
Education level					
High school and under	54 (68.4)	25 (31.6)	4.48	0.034	0.337
University and higher	67 (82.7)	14 (17.3)			
Occupation					
Unemployed	29 (60.4)	19 (39.6)	8.60	0.003	0.232
Employed	92 (82.1)	20 (17.9)			
Monthly income (10,000 KRW)					
<200	22 (53.7)	19 (46.3)	14.4	<0.001	0.300
≥200	99 (83.2)	20 (16.8)			
Diagnosis					
Rheumatoid arthritis	46 (75.4)	15 (24.6)	4.17	0.124	0.161
Systemic lupus erythematosus	29 (65.9)	15 (34.1)			
Ankylosing Spondylitis	46 (83.6)	9 (16.4)			
Duration since diagnosis (year)					
<10	54 (83.1)	11 (16.9)	3.30	0.069	0.144
≥10	67 (70.5)	28 (29.5)			
Perceived health status					
Poor	11 (52.4)	10 (47.6)	8.02	0.018	0.224
Moderate	49 (75.4)	16 (24.6)			
Good	61 (82.4)	13 (17.6)			
Perceived social isolation (VAS), Mean ± SD	3.68 ± 3.15	6.77 ± 2.90	−5.43	<0.001	1.000

Values are frequencies (row percentage); KRW = Korean won; VAS = visual analogue scale. The effect size is Cohen’s d for the t-test and Cramer’s V for the chi-square.

**Table 3 healthcare-10-01758-t003:** Risk factors of depression by multivariate logistic analysis.

Variables	OR	95% CI	*p*
Age (year)	0.99	0.94–1.03	0.537
Education level			
High school and under	2.35	0.70–7.93	0.168
University and higher	reference		
Occupation			
Unemployed	3.61	1.02–12.79	0.047
Employed	reference		
Monthly income (10,000 KRW)			
<200	3.92	1.04–14.80	0.044
≥200	reference		
Perceived health status			
Poor	16.93	1.93–148.18	0.011
Moderate	11.78	1.51–91.98	0.019
Good	reference		
Perceived social isolation (VAS)	1.17	0.95–1.44	0.137
PTSD (IES-R)	1.10	1.05–1.16	<0.001
Insomnia (ISI)	1.26	1.11–1.42	<0.001
Likelihood Ratio	Chi-Square = 94.01	*p* < 0.001	
R-Square	0.44		
Max-rescaled R-Square	0.66		
Hosmer and Lemeshow Goodness-of-Fit	Chi-Square = 12.55	*p* = 0.128	

CI: confidence interval; OR: odds ratio; KRW = Korean won; VAS = visual analogue scale; PTSD = post-traumatic stress disorder; IES-R = impact of events scale-revised; ISI = insomnia severity index.

## Data Availability

Data are available upon request.

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
