# Peer review of "Prevalence and Risk Factors of Depression in Patients with Rheumatic Disease in South Korea during the COVID-19 Pandemic"

_healthcare, 2022, doi:10.3390/healthcare10091758_

Round 1

Reviewer 1 Report (Previous Reviewer 1)

Authors included most of my suggestios in the revised version of the manuscript. I recommend its publication.

Reviewer 2 Report (Previous Reviewer 2)

Thank you for addressing my comments on the initial draft of this manuscript. I have no additional comments at this time.

This manuscript is a resubmission of an earlier submission. The following is a list of the peer review reports and author responses from that submission.

Round 1

Reviewer 1 Report

I want to thank the authors and the Editorial Board for the opportunity to review the article submitted to the Healthcare. The authors’ manuscript refers to a very important topic: depression in rheumatic disease patients during the COVID-19 pandemic.

The presented study has been reported correctly and presents interesting and important results from a practical point of view. Unfortunately, I believe that the results section and discussion section should be reworked. In Table 2 and the description of its results, the authors refer only to the category of statistical significance. As the value of the p-value is strongly dependent on the size of the studied sample (see Lakens, 2022), instead of discussing the significance of the obtained results, I encourage the authors to discuss the size of the observed effects.

Therefore, I recommend that Table 2 should be supplemented with additional effect size measures (e.g. Cohen’s d for the t-test family and Cramer’s V and/or Yule’s Phi for the chi-squared family). From the practical point of view, it is extremely important to discuss the effect size measures. I will ask a rhetorical question: should significant but very small (marginal) effects are considered at all? Based on my recommendation, the discussion section should also be reworked based on the additional effect size measures.

In summary, I recommend the authors’ manuscript for publication in Healthcare after some minor revisions.

Author Response

Manuscript ID: healthcare-1812807

*Title: Prevalence and risk factors of depression in patients with rheumatic disease in South Korea during the COVID-19 pandemic.

We appreciate the time and effort that you and the reviewers have dedicated to providing your valuable feedback on my manuscript. We are grateful to the reviewers for their insightful comments on my paper. We have been able to incorporate changes to reflect most of the suggestions provided by the reviewers. We have highlighted the changes within the manuscript.

Here is a point-by-point response to the reviewers’ comments and concerns.

Point-to-point Responses to the Reviewer's Comments

# Reviewer 1

The presented study has been reported correctly and presents interesting and important results from a practical point of view. Unfortunately, I believe that the results section and discussion section should be reworked. In Table 2 and the description of its results, the authors refer only to the category of statistical significance. As the value of the p-value is strongly dependent on the size of the studied sample (see Lakens, 2022), instead of discussing the significance of the obtained results, I encourage the authors to discuss the size of the observed effects.

Therefore, I recommend that Table 2 should be supplemented with additional effect size measures (e.g. Cohen’s d for the t-test family and Cramer’s V and/or Yule’s Phi for the chi-squared family). From the practical point of view, it is extremely important to discuss the effect size measures. I will ask a rhetorical question: should significant but very small (marginal) effects are considered at all? Based on my recommendation, the discussion section should also be reworked based on the additional effect size measures.

Response:

Thank you for your great comment.

LINE 125,141-142, 213-219, Table 2. Revisions have been made according to your comments.

Reviewer 2 Report

This study examined factors associated with depression among patients with rheumatic diseases in the context of the COVID-19 pandemic. Strengths of this manuscript include clear description of the methods and the timeliness of the subject. Overall, this is a simple and well-executed study, but I have some comments that I think could be addressed to enhance its impact.

  1. Elements of national response to the pandemic may have contributed to, or mitigated, the experience of isolation and depression (e.g., lockdowns and social distancing protocols). Because these responses varied greatly between countries, and given the international nature of the readership of this journal, additional context on measures taken specifically in South Korea, both prior to and at the time of data collection, would be helpful to add to the introduction.

  2. The discussion could be improved by expanding on the potential applications of these findings. How can an understanding of depression in this population impact public health policy or clinical decision making?

Author Response

Manuscript ID: healthcare-1812807

*Title: Prevalence and risk factors of depression in patients with rheumatic disease in South Korea during the COVID-19 pandemic.

We appreciate the time and effort that you and the reviewers have dedicated to providing your valuable feedback on my manuscript. We are grateful to the reviewers for their insightful comments on my paper. We have been able to incorporate changes to reflect most of the suggestions provided by the reviewers. We have highlighted the changes within the manuscript.

Here is a point-by-point response to the reviewers’ comments and concerns.

Point-to-point Responses to the Reviewer's Comments

# Reviewer 2

  1. Elements of national response to the pandemic may have contributed to, or mitigated, the experience of isolation and depression (e.g., lockdowns and social distancing protocols). Because these responses varied greatly between countries, and given the international nature of the readership of this journal, additional context on measures taken specifically in South Korea, both prior to and at the time of data collection, would be helpful to add to the introduction.

Response:

Thank you for your great comment.

LINE 42-47. Revisions have been made according to your comments.

  1. The discussion could be improved by expanding on the potential applications of these findings. How can an understanding of depression in this population impact public health policy or clinical decision making?

Response:

Thank you for your great comment.

LINE 295-299. Revisions have been made according to your comments.
